# Reliable Predictors of Muscle-Invasive Upper Tract Urothelial Carcinoma before Nephroureterectomy: Why, to Whom, and How Should We Perform Lymph Node Dissection?

**Julian Chavarriaga** [1,2,*]**, Juan Erazo** [1]**, Lupi Mendoza** [1]**, German Ramirez** [1]**, Jorge Sejnaui** [1] **and Carlos Morales** [1]

[1]  Division of Urology, Clinica Imbanaco, Quiron Salud, Carrera 38 Bis No. 5B2-04, Cali 760001, Colombia; juan.erazo@imbanaco.com.co (J.E.); Lupi.mendoza@imbanaco.com.co (L.M.); german.ramirez@imbanaco.com.co (G.R.); Jorge.sejnaui@imbanaco.com.co (J.S.); carlos.morales@imbanaco.com.co (C.M.)
[2]  Division of Urology, Pontificia Universidad Javeriana, Bogotá 110231, Colombia
*   Correspondence: chavarriagaj@javeriana.edu.co; Tel.: +57-(1)-5946107

**Abstract:** (1) Introduction and Objective: Upper tract urothelial carcinoma (UTUC) is an uncommon disease, only accounting for 5–10% of all urothelial carcinomas. Current clinical practice guidelines encourage a risk-adapted approach to UTUC management, including lymph node dissection (LND) in patients with muscle-invasive or high-risk tumors. If pathological characteristics could be more accurately predicted from preoperative data, we could optimize perioperative management strategies and outcomes. The aim of this article is to present a detailed revision of preoperative predictors for muscle-invasive UTUC, locally advanced or advanced UTUC, as well as current indications, technique variations, and the reasons as to why LND should be offered to these patients. (2) Methods: We included any kind of studies related to information concerning UTUC, nephroureterectomy, LND, risk factors for recurrence, prediction tools and models for risk stratification. A literature search was conducted following medical subject headings (MeSh), Emtree language, Decs, and text words related. We searched through MEDLINE (OVID), EMBASE (Scopus), LILACS, and the Cochrane Central Register of Controlled Trials (CENTRAL) from inception to May 2021. Evidence acquisition was presented according to the PRISMA diagram. (3) Results: Preoperative risk factors for either muscle-invasive UTUC ($\geq$pT2), extra urothelial recurrence (EUR), locally advanced disease, or high-risk UTUC can either be derived from ureteroscopic (URS) findings, urine cytology, URS biopsy, or from preoperative radiologic findings. It seems reasonable that LND may provide not only staging and prognostic information but also play a therapeutic role in selected UTUC patients. The patients who benefit the most from LND appear to be those with $\geq$ pT2 disease, because patients with tumors $\leq$ pT1 rarely metastasized to LNs. UTUC has characteristic patterns of lymphatic spread that are dependent on tumor laterality and anatomical location. Choosing the right patients for LND, designing and standardizing LND templates based on tumor location and laterality is critical to improve LN yield, survival outcomes, and to avoid under-staging or overtreatment. (4) Conclusions: Patients with muscle-invasive or non-organ-confined UTUC have an extremely high risk for disease recurrence and cancer-specific mortality (CSM). Preoperative factors and prediction models must be included in the UTUC management pathway in our clinical practice to improve the accurate determination of high-risk groups that would benefit from LND. We recommend offering LND to patients with ipsilateral hydronephrosis, cHG, cT1 at URS biopsy and renal sinus fat or periureteric fat invasion. The role of lymphadenectomy in conjunction with radical nephroureterectomy (RNU) is still controversial, given that it may result in overtreatment of patients with pTa-pT1 tumors. However, a clear benefit in terms of recurrence-free survival (RFS) and cancer-specific survival (CSS) has been reported in patients with $\geq$pT2. We try to avoid LND in patients with cLG, cTa, and no ipsilateral hydronephrosis if the patient is expected to be compliant with the follow up schedule. There is still plenty of work to do in this area, and new molecular and non-invasive tests are necessary to improve risk stratification.

**Keywords:** carcinoma; transitional cell; kidney; pelvis; renal; ureteral neoplasm; lymph node excision; ureteroscopic surgeries

## 1. Introduction

Upper tract urothelial carcinoma (UTUC) is an uncommon disease, only accounting for 5–10% of all urothelial carcinomas [1–3]. The current standard of care for non-metastatic UTUC is radical nephroureterectomy (RNU) with the excision of a bladder cuff. Unfortunately, UTUC is a biologically aggressive tumor with a high potential for disease recurrence, metastasis, and cancer-specific mortality (CSM) [4,5]. Trying to overcome this issue, current clinical practice guidelines encourage a risk-adapted approach to UTUC management. Including a detailed preoperative tumor invasiveness assessment, and recommending either neoadjuvant chemotherapy (NAC) or lymph node dissection (LND) in patients with muscle-invasive or high-risk tumors, which includes patients with clinical high grade (cHG) when the diagnosis is made by means of cytology or ureteroscopic biopsy and pathological high grade (pHG) when the diagnosis has been made by radical or nephron-sparing surgery [6]. Due to the low incidence of UTUC, the majority of studies mainly consist of single-institution studies, resulting in low-level evidence for most recommendations [1,2,6–9].

If pathological characteristics could be more accurately predicted from preoperative data, we could optimize perioperative management strategies and outcomes. Beyond the established prognostic factors such as clinical stage (cT), high grade ureteroscopic (URS) biopsy, and positive urine cytology, another subset of factors may play an important role in predicting the need for LND, such as lymphovascular invasion (LVI), hydronephrosis, length of the ureteral tumor, tumor location, glomerular filtration rate (GFR), and neutrophil counts [1,2,6–9]. Recent efforts to combine imaging and ureteroscopic variables to accurately identify ≥ pT2 disease have been made. The integration of these factors in predictive tools or models is gaining acceptance to guide decision-making for personalized care delivery [2].

Because of the difficulty in accurately choosing patients for LND in clinically node-negative UTUC, it has been recommended that a LND should be performed in all patients [10]. We are convinced that there is a subset of patients with UTUC that will benefit from this staging procedure and another subset of patients for whom LND can be safely omitted, avoiding surgery-related complications. The aim of this article is to present a detailed revision on preoperative predictors for muscle-invasive or locally advanced UTUC, as well as analyzing current indications, technique variations, and the reasons why a LND should be offered to these patients.

## 2. Methods

We included any kind of study related to information concerning UTUC, nephroureterectomy, LND, risk factors for recurrence, prediction tools, and prediction models for risk stratification. We included reviews, systematic reviews, and primary studies. The literature search was conducted in accordance with the use of medical subject headings (MeSh), Emtree language, Decs, and text words related. We searched through MEDLINE (OVID), EMBASE (Scopus), LILACS, and the Cochrane Central Register of Controlled Trials (CENTRAL) from inception to May 2021. To ensure literature saturation, we scanned references from relevant articles identified through the search, conferences, thesis databases, Open Grey, Google scholar, and clinicaltrials.gov. We contacted authors by e-mail when there was missing information.

A grey literature search was also performed on the pages of The National Technical Information Service (NTIS) and the European Association for Grey Literature Exploitation (EAGLE); however, no additional relevant information was found.

The search criteria were established in the form of free text and indexed terms. To characterize UTUC, we used the free terms: "upper tract urothelial carcinoma", "renal pelvis", "transitional cell carcinoma", "ureteral carcinoma", "prediction tools", "prediction models", "retroperitoneal lymph node dissection", "muscle invasive", "UTUC muscle-invasive", and we used the following MeSH terms "Lymph node dissection", "Transitional cell carcinoma", "Ureteral neoplasm", "Renal pelvis", "Local neoplasm recurrence". The search was limited to publications in the last 10 years, and we used the following search strategy: "Lymph Node Excision" [Mesh] AND "Carcinoma, Transitional Cell" [Mesh] AND "Ureteral Neoplasms" [Mesh] AND "Kidney Pelvis" [Mesh].

Four researchers reviewed each reference by title and abstract. Then, we scanned full texts of relevant studies. We included papers reporting data on predictive factors or models for muscle-invasive UTUC, and studies reporting on LND limited to UTUC. We excluded studies that did not fulfill the abovementioned criteria and that were written in a language different from English. Duplicate studies were removed. Evidence acquisition is presented according to the PRISMA diagram (Figure 1).

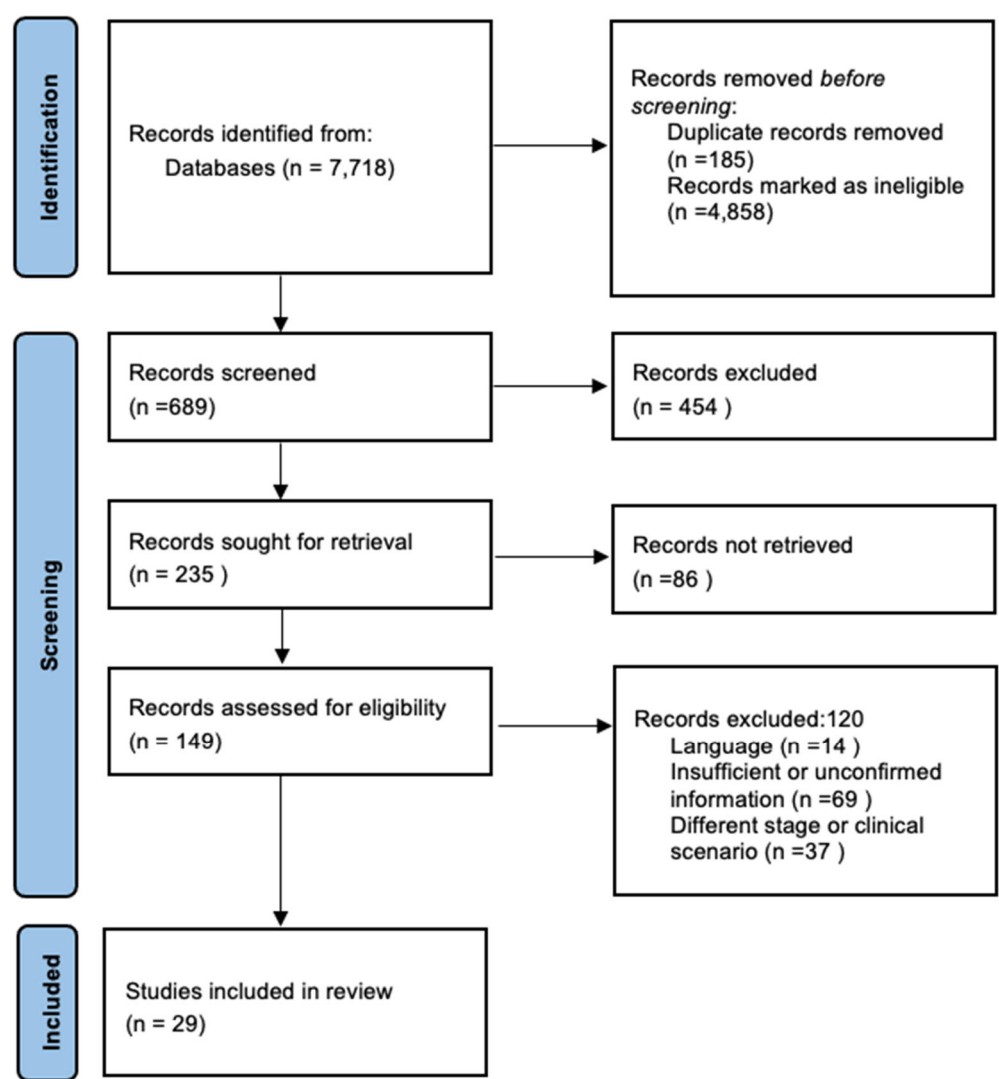

**Figure 1.** Evidence acquisition according to the PRISMA diagram.

## 3. Discussion

### 3.1. Predictors of Muscle-Invasive UTUC

Preoperative risk factors for either muscle-invasive UTUC ($\geq$pT2), extra urothelial recurrence (EUR), locally advanced disease, or high-risk UTUC have been described [1–3,5,8,9].

These factors can either be derived from URS findings, cytology, URS biopsy, or from pre-operative radiographic findings. Most of these tests are included in routine preoperative evaluation for suspected UTUC.

Discordance between URS biopsy and final surgical pathology is high, and the major challenge is that the use of small caliber biopsy forceps or baskets normally yields small fragments of tissue, which increase the difficulty of accurately establishing tumor clinical grade and stage. Clinical staging by radiographic characteristics has its limitations; it can be difficult to predict tumor stage in UTUC, mainly because of different tumor locations and characteristics. Prior research showed that cross-sectional imaging has an accuracy of only 52%, tending to overstage most patients, especially when hydronephrosis was present, which was found in 80% of overstaged cases [11].

### 3.1.1. Ureteroscopic Predictive Factors

Margolin et al. [12] evaluated 314 patients with UTUC who had undergone URS biopsy and subsequently RNU to determine pathology discordance. They reported that on URS biopsy, 61% had cHG tumors and 21% had subepithelial connective tissue invasion (cT1+). As expected in RNU pathology, 79% had pHG tumors and 45% had stage $\geq$pT2. Urine cytology was collected and analyzed in 230 cases, and it was positive only in 37% of the cases. The probability of missing invasion (cT1+) when URS was performed was significantly increased when biopsy fragments were $\leq$1 mm, and using forceps was associated with a higher likelihood of identifying smaller fragments. Only three preoperative factors for $\geq$pT2 UTUC were statistically significant in multivariate analysis: cHG (OR 2.4, 95% CI 1.1–5.2, $p = 0.04$), cT1+ (OR 9.0, 95% CI 3.2–25.6, $p < 0.001$), and advanced age (OR 1.0, 95% CI 1.0–1.1, $p = 0.02$). cHG combined with cT1+ reached a positive predictive value (PPV) of 86% [12].

Few studies have assessed predictors of recurrence in ureteral carcinoma [5]. Ito et al. conducted a retrospective study including 70 patients, and they found that 30% developed EUR, and 66% in regional lymph nodes [5]. EUR-free survival was significantly worse in patients with $\geq$pT3 disease (HR 7.69), length of ureteral cancer along the ureter $\geq$ 3 cm (HR 3.90), positive cytology (HR 4.90), eGFR < 60 mL/min/1.73 m$^2$ (HR 6.57), maximal diameter of ureteral cancer $\geq$ 1.6 cm, and neurophil-to-lymphocyte ratio (NLR) > 3.0. They stratified patients into three risk groups, according to the number of risk factors present (0, 1–2, and $\geq$3), a 3-year EUR-free survival of 100%, 81.4%, and 25.1% was found according to the aforementioned risk groups, respectively. One of the main limitations of this study was that not all patients underwent LND, they did not use a template-based LND, and adjuvant chemotherapy was administered to patients with $\geq$pT2 disease, which could have altered the rate of EUR [5].

### 3.1.2. Radiographic Predictive Characteristics

Optimal preoperative radiographic staging is required to appropriately tailor surgical treatment for individual patients. Hydronephrosis on preoperative axial computed tomography (CT) has been associated with features of high-risk UTUC and is a good predictor of advanced stage ($\geq$pT2) for both, renal pelvis and ureteral tumors [8].

Messer J. et al., in a study of 469 patients, evaluated whether ipsilateral hydronephrosis was a reliable predictor of advanced pathological stage—55% had preoperative ipsilateral hydronephrosis. At final pathology revision, 47% had $\geq$pT2 disease, 36% non-organ-confined disease ($\geq$pT3 and/or pathological positive nodes (pN+)) and 73% had high grade UTUC. Hydronephrosis was a statistically significant predictor of $\geq$pT2 disease (HR 7.4, $p < 0.001$), $\geq$pT3, pN+ (HR 5.5, $p < 0.001$), and pHG (HR 1.6, $p < 0.03$) [8]. The conclusion was that the simplest radiographic predictive factor to determine muscle-invasive, non-organ confined disease or pHG was hydronephrosis [8]. These findings are in contrast to a previous study, in which hydronephrosis tended to overstage 80% of the tumors [11]. This study was conducted between 1984 and 1995 and included 31 patients. The technological improvements in the cross-sectional imaging techniques in the upcoming years, as well

as the limited number of patients, might explain the contradictory findings regarding Messer's et al. study.

### 3.1.3. Combined Radiographic and Ureteroscopic Variables

Attempts at combining radiographic characteristics with URS biopsy and urine cytology have been made in order to improve prediction of ≥pT2 stage UTUC. Brien et al. reported a study of 172 patients, 54% with ipsilateral hydronephrosis, 43% with cHG on URS biopsy, and 80% with a positive cytology. On multivariate analysis, ipsilateral hydronephrosis was a significant predictor of pT2 (HR 12.0, $p < 0.001$), non-organ confined (HR 5.2, $p < 0.001$), and high grade (HR 2.3, $p = 0.04$). Positive cytology was a predictor of non-organ confined UTUC (HR 3.1, $p = 0.035$) instead of pT2 or pHG. URS biopsy grade was associated with pT2 (HR 4.5, $p < 0.001$), non-organ confined (HR 3.9, $p < 0.001$), and pHG (HR 25.9, $p < 0.001$) [9]. When combining all three variables into a single model (hydronephrosis, cHG, and positive cytology), they found 89% PPV for pT2 stage and 73% for non-organ confined UTUC. A negative predictive value (NPV) of 100% was reported when all three variables were normal [9].

Chen XP et al. analyzed data of 729 patients. They found in multivariate analysis that male gender (hazard ratio (HR 1.898, $p = 0.001$), sessile architecture (HR 3.249, $p < 0.001$), cHG (HR 5.007, $p < 0.001$), ipsilateral hydronephrosis (HR 4.768, $p < 0.001$), renal pelvis UTUC (HR 2.620, $p < 0.001$), and no multifocality (HR 1.639, $p = 0.028$) were significant predictors for muscle-invasive UTUC [1]. The reported accuracy was 79% for both pT2 stage and non-organ confined UTUC.

Favaretto et al. aimed to create a preoperative model to identify patients at risk of ≥pT2 stage and non-organ confined UTUC. They retrospectively analyzed data from 274 patients treated with RNU. Overall, 49% had ≥pT2 stage and 30% had non-organ confined UTUC at final pathology. In the univariate analysis, local invasion in imaging (defined as renal sinus fat or periureteric fat) ($p < 0.001$), hydronephrosis ($p = 0.011$), and URS biopsy cHG tumors were all significantly associated with increased risk of ≥pT2 stage, and all variables, with the exception of hydronephrosis, were also significantly associated with non-organ confined UTUC. Tumor location and hydronephrosis were not significant predictors of ≥pT2 stage or non-organ confined UTUC at the multivariable models ($p = 0.6$ and $p = 0.7$), ($p = 0.065$ and $p = 0.4$), respectively. However, cHG ($p = 0,04$ and $p = 0,005$) and local invasion on imaging ($p = 0.017$ and $p = 0.001$) were both significantly associated with ≥pT2 stage and non-organ confined UTUC. The accuracy to predict ≥pT2 and non-organ confined UTUC was 71% and 70%, respectively [2].

### 3.2. The Rationale of Lymph Node Dissection in UTUC
#### 3.2.1. Why?

LN metastases in UTUC are common, with a reported incidence of up to 30–40% [13–17]. LN dissemination typically precedes the identification of visceral metastases [13–15,18]. It seems reasonable that LND may provide not only staging and prognostic information, but also plays a therapeutic role in selected UTUC patients [14,16,17]. It is clear that pN+ patients have a significantly worse prognosis when compared to pN0 patients [13–17]. Therefore, control of nodal metastases is an important milestone in the treatment of this disease.

Nodal status has been thought to be a significant predictor of CSS in UTUC. Roscigno et al. examined data of 1130 patients treated with RNU and LND. Overall, they found that 5-year CSS was lower in patients with pN+ compared with pNX (35% vs. 69%, $p < 0.001$), and in pNx vs. pN0 (69% vs. 77%, $p = 0.024$). In a sub-analysis of patients with pT2-4 disease, CSS rates were lowest in pN+ followed by pNx and pN0 (33% vs. 58% vs. 70%, $p < 0.017$). These findings suggest that pNx was significantly associated with worse prognosis than pN0 in pT2-4 UTUC, which corroborates the suggestion that patients expected to have pT2-4 disease should undergo LND [17].

In a recent meta-analysis, LND failed to show a statistically significant effect on CSS (HR 1.17, 95% CI: 0.93–1.48, $p = 0.18$). Patients with pN0 did not have a better

CSS compared with those with pNx (HR 0.99, 95% CI: 0.81–1.22, *p* = 0.95); as expected, significant heterogeneity (I2 = 94%, Chi2 = 35.97, *p* < 0.00001) was found. LND had no association with better recurrence free survival (RFS), the pooled HR for RFS was 1.33 (95% CI: 0.87–2.06, *p* = 0.19) [16]. These findings are in contrast with a systematic review by the European Association of Urology Guidelines Panel on non–muscle-invasive bladder cancer, evaluating the potential benefit of LND during RNU for UTUC [15]. Dominguez-Escrig et al. demonstrated a survival benefit with LND in ≥pT2 disease of the renal pelvis [15].

A recent population-based analysis of trends in LND for UTUC, using the surveillance epidemiology and end results (SEER) database, showed a significant increase in LND rates at the time of RNU from 20% in 2004 to 33% in 2012, and a tendency towards improved 5-year CSS was observed in the highest quartile of LNs harvested [19]. One of the main concerns of LND in UTUC is that there is huge variation in surgical practices, even at the same institutions. Strategies for standardizing the procedure, selecting the appropriate candidates, and adopting surgical adequacy parameters are required in order to improve oncological outcomes.

### 3.2.2. To Whom

As previously reported, the patients who benefit the most from LND appear to be those with ≥pT2 disease. Abe et al. suggested that the incidence of LN involvement is 1.9%, 4.5%, 8.9%, 28.7% and 70.1% for pTa, pT1, pT2, pT3, and pT4, respectively [20]. Miyao et al. reported slightly higher rates of LN metastasis in cT1 (13.3%) [20].

In patients with node-negative disease (cN0), Kondo et al. reported that 42.5% of tumors ≤cT1 are upstaged to a clinical stage ≥pT2 and would lose the benefit of LND, if LND is omitted at the time of RNU. The latter makes it difficult to trust conventional staging to select candidates for LND based only on clinical stage [20]. Therefore, many experts in the field, as well as EAU clinical practice guidelines recommended LND in all patients who undergo RNU, except for those with severe comorbidities.

Lughezzani et al. performed a population-based analysis using the SEER database, including 2824 patients treated with RNU between 1988 and 2004. They reported a 5-year CSS of 81.2% for pN0 and 77.8% for pNx. No association was found between pNx and pN0 status with CSS using univariable and multivariate analysis [13]. These findings suggest that there is no survival benefit related to the performance of LND in UTUC [13]. However, Brausi et al., in a retrospective study including 82 patients with ≥pT2 UTUC, compared patients who underwent LND versus those who did not, in terms of oncological outcomes. A total of 40 patients (48.8%) underwent LND. Of the patients who underwent LND, 40% were pN+ and 60% were pN0. The group of patients that did not undergo LND were classified as pNX and 54% of these patients developed recurrences or progressed. RFS and CSS rates were 64.3% and 81.6% in patients treated with LND, respectively, and alarmingly, in the group of patients in which LND was not performed, RFS and CSS rates were 46.3% and 44.8%, showing a significant benefit of LND in ≥pT2 disease (RFS, *p* = 0.03; CSS, *p* = 0.007) [21].

The major pitfall of LND decision-making is that it remains difficult to reliably identify patients with ≥pT2 disease before RNU. Many experts continue to recommend performing LND in all patients treated with RNU if accurate staging is desired. However, this approach could lead to overtreating some patients [11–16]. We believe that all patients with suspicion of ≥pT2 disease must undergo LND at the same time of RNU, regardless of the surgical approach (laparoscopic, robotic or open). For patients with no clear landmarks of muscle-invasive or locally advanced disease, we recommend using the following predictive models or predictive variables that have shown significant predictive capabilities for ≥pT2 disease:

- Ipsilateral hydronephrosis + cHG + positive urine cytology (The three variables combined have 89% PPV and 100% NPV) [9] (Figure 2)
- cHG + cT1 at URS biopsy (PPV 86%) [12]
- Sessile architecture, cHG, ipsilateral hydronephrosis, renal pelvis UTUC, and no multifocality (79% accuracy) [1] (Figure 2)

- Local invasion on imaging (renal sinus fat or periureteric invasion), ipsilateral hydronephrosis, and cHG (71% accuracy) [2] (Figure 2)

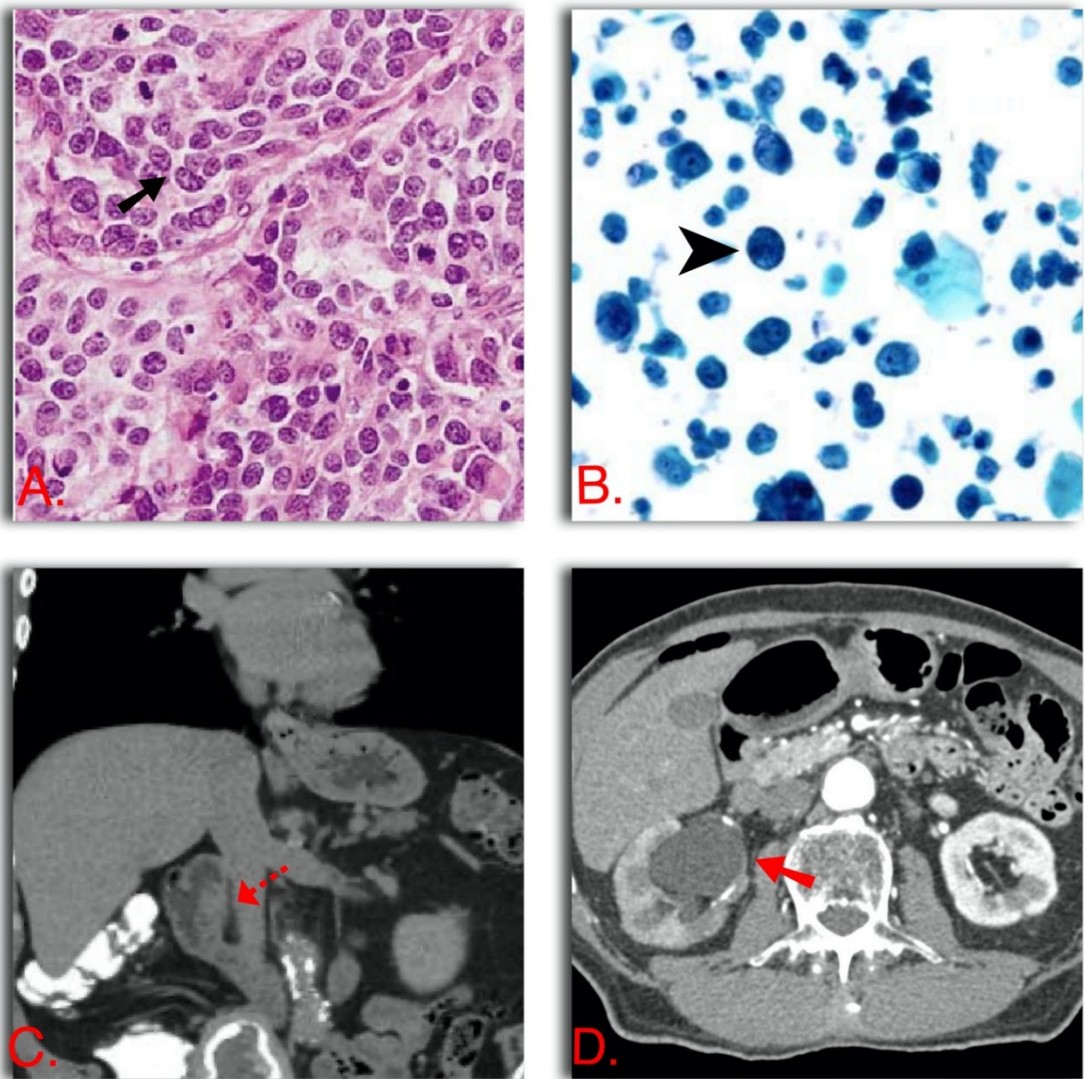

**Figure 2.** Predictive factors for ≥pT2 UTUC. (**A**) Hematoxylin-eosin 20× cHG transitional cell carcinoma, diagnosed with URS biopsy of a renal pelvis tumor. (**B**) Positive urinary cytology (in this case HG). (**C**) Local invasion of UTUC (periureteric fat extension). (**D**) Right hydronephrosis secondary to proximal ureteral transitional cell carcinoma.

We recommend performing LND at the time of RNU if a patient has muscle-invasive disease or in a patient with any of the following characteristics: ipsilateral hydronephrosis, cHG, cT1 at URS biopsy or radiographic signs of renal sinus fat or periureteric fat invasion. If any of the variables are present at preoperative evaluation, the risk of losing the potential benefit of LND is too high; therefore, all these patients should undergo LND.

### 3.2.3. How

Anatomical-Based Templates for LND

UTUC has characteristic patterns of lymphatic spread that are dependent on tumor laterality and anatomical location. Adopting LND templates based on tumor location and laterality is critical to improve LN yield, survival outcomes, and to avoid under staging or overtreatment. Kondo et al. reported a detailed anatomical mapping study of regional LNs in patients with pN+ UTUC. This study showed that right-sided renal pelvis, proximal and

mid ureteral tumors encompassed a wider area for nodal dissemination than previously thought, including the renal hilum, paracaval, retrocaval, and interaortocaval regions. Left-sided renal pelvis, proximal, and mid ureteral tumors always metastasized to para-aortic and left renal hilum nodes and distal ureteral tumors, regardless of the laterality metastasized to intrapelvic LNs [10] (Figure 3).

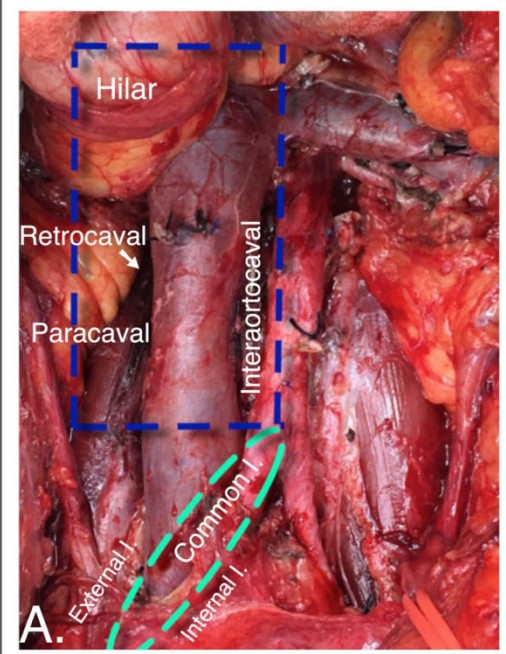 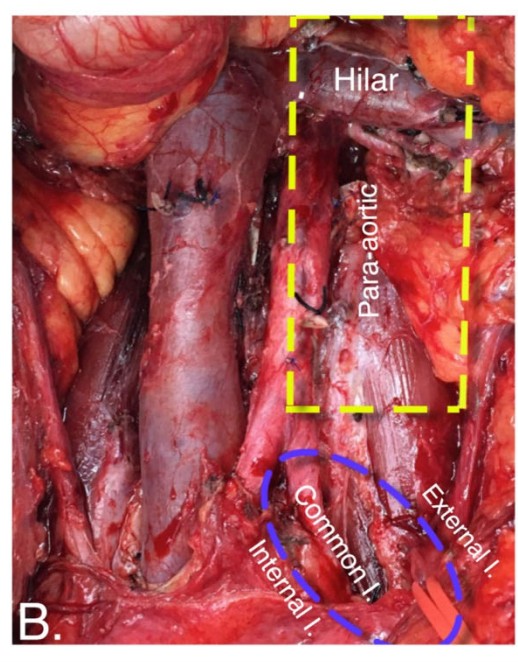

**Figure 3.** Regional LND templates for anatomical LND in UTUC. (**A**) Blue template for a renal pelvis or proximal or mid ureteral right-sided UTUC. Green template for distal or mid ureteral right-sided UTUC. (**B**) Yellow template for a renal pelvis or proximal or mid ureteral left-sided UTUC. Purple template for distal or mid ureteral left-sided UTUC.

Matin et al. conducted a retrospective multicentric study including 73 patients with non-metastatic pN+ UTUC to determine the patterns of lymphatic metastases and aiming to propose consolidated templates for LND. They reported that right-sided renal pelvis tumors had LN metastases to the hilum, paracaval, retrocaval and interaortocaval regions in 22.1%, 44.1%, 10.3%, and 20.6% of cases, respectively. Left-sided renal pelvis tumors metastasized to the hilum in 53% of the cases, with paraaortic, interaortocaval, common iliac, and aortic bifurcation in 31%, 4%, 1%, and 1%, respectively. The right-sided proximal ureteral tumors had lymph node metastases to the hilum in 46.2% of cases, and to paracaval and retrocaval regions in 46.2% and 7.7%, respectively. In a similar fashion, in the left-sided proximal ureteral tumors, LN metastases were found in 36.4% of hilar nodes and 63.6% of paraaortic nodes. The four patients with distal ureter tumors had LN metastases to the para-aortic, common iliac, external iliac, and internal iliac stations in 33.3%, 33.3%, 16.7%, and 16.7% of cases, respectively [18] (Figure 3).

Template-based regional LND plays an important role in reducing locoregional recurrence in patients with cN0 UTUC, as shown by Matsumoto et al. [22]. They evaluated 105 patients with cTa-3N0M0 UTUC who were treated with regional LND during laparoscopic RNU, using the anatomical-based templates proposed by Abe et al. [22]. In renal pelvis or proximal ureteral tumors (considered above as the crossing of the common iliac artery), renal hilar, paracaval, retrocaval, and interaortocaval LNs were removed if the tumor was located on the right side, and if it was a left-sided tumor, renal hilar and para-aortic lymph nodes were harvested. In patients with tumors of the distal ureter, ipsilateral obturator and common, external, and internal iliac nodes were harvested. The median number of LN harvested was 12, and seven patients (6.7%) had LN metastases on hematoxylin-eosin. Immunohistochemistry (IHC) revealed micrometastases in five addi-

tional patients. One of the highlights of this paper is that of the seven pN+ patients, 71% developed distant organ or nodal metastases, and of the five pN+ micrometastatic patients, only one (20%) developed distant metastases, suggesting that regional LND could promote local disease control by eliminating micrometastasis in patients previously thought to be cN(−). Five-year CSS was 95% for pN0, 53.3% for pN+ (Detected with IHC), and 23.8% for pN+ [22]. These data support the use of an anatomical template-based LND for UTUC and confirms that CSS is determined by nodal status and that LND could play an important role improving survival [22].

Kondo et al. also suggested a potential benefit of template-based LND, and they demonstrated that UTUC patients in which LND was performed had better CSS than those who did not undergo LND or had a suboptimal LND ($p = 0.06$). This benefit was even better in patients with ≥pT3 disease ($p = 0.01$). CSS was likely to improve when the number of lymph nodes removed increased, especially when seven or more nodes were harvested [23].

Rao et al. proposed a modified LND template. In patients with right-sided UTUC, the template included hilar, paracaval, and right pelvic LNs. In patients with left-sided UTUC, the template included perihilar, para-aortic, and left pelvic LNs. The mean LNY was seven. None of the patients died, 85% of them were disease free, and 10% had evidence of metastatic disease. The morbidity of the LND was acceptable, mostly with minor complications (Clavien grades I and II) [24].

We recommend performing anatomical template-based LND according to tumor laterality and location. It certainly has been shown not only to improve the LNY but also avoids over-dissecting areas in which lymphatic spread does not occur. We used the anatomical-based templates proposed by Abe et al. (Figure 3). We believe with these templates, all the regions in which LN metastasis could be found are dissected and the nodes are removed. One question that remains unanswered is: should we offer LND after RNU when LND was not performed initially? We consider this to be a difficult question that may have more than one correct answer. If we are trying to improve CSS in patients with ≥pT2 UTUC, we should offer LND after RNU, a thorough discussion of the potential risks and side effects with the patient is mandatory, before making the decision. On the other hand, some experts could argue that with the results of the POUT trial, all patients staged as pT2–T4 pN0 may benefit from cisplatin or carboplatin-based adjuvant chemotherapy, and that LND at a second stage could be avoided [24].

Lymph Node Yield

The number of lymph nodes removed has significant prognostic value for many malignancies, including UTUC [25]. A higher lymph node yield (LNY) is significantly associated not only with finding positive nodes but also increasing the probability of finding multiple positive nodes. Surgeon volume is the most important factor to account for LNY, and open surgery has been proposed as another factor increasing LNY. Thompson et al. in a study of 124 patients treated with RPLND for testicular cancer, reported a mean total node count of 51, and mean node counts for the paracaval, interaortocaval, and paraaortic regions of 8, 17, and 26 nodes, respectively. They concluded that these results should provide a benchmark for surgical adequacy, even in UTUC. For example, if a paracaval LND is performed, at least eight lymph nodes on average should be counted, and if a paraaortic LND is performed for a left tumor, 26 lymph nodes on average should be harvested [25]. These findings are important, although it is quite difficult to extrapolate parameters for surgical adequacy from testicular cancer to UTUC, given that the two are very different diseases, with different landing pathways and zones in the retroperitoneal LNs.

Roscigno et al. in a multicentric study including 551 patients with UTUC, examined the minimal number of LNs that needed to be harvested to certainly detect LN involvement. They reported that a total of 13 LNs needed to be harvested in order to achieve a 90% probability of detecting metastases, and the removal of eight or more LNs was associated with a 75% probability of detecting one or more positive node [17]. The same author

found that the LNY was not associated with CSM in univariable (HR 0.99; $p = 0.16$) or in multivariable (HR 0.97; $p = 0.12$) analyses. However, in the subgroup of pN0 patients, LNY achieved the independent predictor status of CSM (HR 0.93; $p = 0.02$). They concluded that a LNY of eight or more was the best variable predicting CSM (HR: 0.42; $p = 0.004$) [26]. With the small amount of evidence on this specific topic, and the very well conducted and designed studies by Roscigno et al. [17,26], we used the LNY of eight or more LNs as a parameter of surgical adequacy in our clinical practice, until novel and better evidence become available. Despite the importance of this parameter, for now, we cannot use them to make any clinical decisions.

Abe et al. found no significant correlation between the LNY and CSS, in a retrospective study including 166 patients that underwent LND for UTUC. The median LNY was six nodes, and when specifically comparing between the 72 patients with <6 LNs removed and the 78 with ≥6 nodes, no significant difference in CSS were found $p = 0.216$. These findings were maintained even after re-categorizing as 6–10 harvested nodes and ≥11 nodes. A limitation of this study is that they did not use a standardized template for LND [27].

A LNY of at least eight LNs has been shown to predict cancer specific mortality and to increase the probability of detecting LN metastases, therefore, arbitrarily in our clinical practice, we have established a threshold of eight or more nodes, as a surrogate of surgical adequacy in LND for UTUC. However, because of the different lymphatic landing pathways, according to tumor location or laterality, and because the templates may vary according to these factors, it is not prudent to set a definitive threshold for this disease.

Lymph Node Density

A useful parameter to predict the risk of recurrence and CSM that has been assessed in many tumors is lymph node density. Defined as the total of positive nodes, divided by the total of harvested nodes, expressed in a percentage. Bolenz et al. suggested that LN density could aid in risk stratification of patients with pN+ UTUC. They conducted a multicentric study including a total of 135 patients with an alarming 4-year recurrence rate of 68% and CSM of 58%. The best LN density threshold to predict recurrence or CSM was ≥30%, with 5-year recurrence and mortality rates of 38% and 48%, respectively. Compared to 25% and 30% when LN density was <30% [28].

Lymph node density for UTUC has only been reported in the MSKCC study, with a proposed threshold of ≥30%, to accurately predict RFS and CSM. At the moment, the only clinical application of using this parameter is to counsel patients. With the recently published POUT trial, all patients staged as pT2–T4 pN0N3 M0 or pTany N1–3 M0, regardless of their LNY or lymph node density, should be offered cisplatin-based chemotherapy or carboplatin in patients unfit for cisplatin within 90 days of surgery [29]. Additionally, the same applies to the most recent study Checkmate 274, in which all patients with urothelial carcinoma (including renal pelvis and ureter) staged as pT3-4, R0, and/or pN+ not eligible for or declined adjuvant chemotherapy, should be offered nivolumab within 120 days of surgery, every 2 weeks for up to 1 year, with a 30% absolute improvement in disease free survival or death [30].

## 4. Conclusions

Patients with muscle-invasive or non-organ-confined UTUC have an extremely high risk for disease recurrence and CSM. Preoperative factors and prediction models must be included in the UTUC management pathway to improve accurate determination of ≥pT2 patients that would benefit from LND. In our own practice, we perform LND in patients with ipsilateral hydronephrosis, cHG, cT1 at URS biopsy, and renal sinus fat or periureteric fat invasion. The role of lymphadenectomy in conjunction with RNU is still controversial, given that it may result in overtreatment of patients with pTa-pT1 tumors. However, a clear benefit in terms of RFS and CSS has been shown in patients with ≥pT2. Experts' opinions and clinical practice guidelines recommend that all patients undergoing RNU should be offer regional LND at the same time, which optimally should be carried out

following template-based anatomical dissections, aiming to harvest the highest possible amount of LNs. Despite the aforementioned recommendation, we still try to avoid LND in patients with cLG, cTa, no ipsilateral hydronephrosis, and no other URS or radiographic signs of non-organ confined disease or muscle-invasive disease, if the patient is expected to be compliant with the follow up schedule. There is still plenty of work to do in this area, and new molecular and non-invasive tests are necessary to improve risk stratification and avoid undertreatment or overtreatment.

**Author Contributions:** Conceptualization, J.C. and J.E.; methodology, J.S., J.C. and C.M.; software, L.M.; validation, G.R., C.M., J.C. and J.E. formal analysis, J.C. and C.M.; investigation, J.C. and C.M.; resources, J.E.; data curation, J.C., G.R. and J.S.; writing—original draft preparation, J.C.; writing—review and editing, C.M.; visualization, G.R., C.M., J.C., J.E., J.S. and L.M.; supervision, C.M. and J.C.; project ad-ministration, J.C.; All authors have read and agreed to the published version of the manuscript.

**Funding:** This research received no external funding.

**Institutional Review Board Statement:** Not applicable.

**Informed Consent Statement:** Written informed consent has been obtained from the patients to publish this paper.

**Conflicts of Interest:** The authors declare no conflict of interest.

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
