# Peer review of "Reliable Predictors of Muscle-Invasive Upper Tract Urothelial Carcinoma before Nephroureterectomy: Why, to Whom, and How Should We Perform Lymph Node Dissection?"

_2673-4397, doi:10.3390/uro1030009_

Round 1

Reviewer 1 Report

This article is designed to inform decision-making on LND in UTUC patients. To achieve this purpose, the article needs significant improvements in the following areas: a) the article in general needs a lot more specifics and details, especially when it comes to recommendation, description of processes, terminology, and acronyms; b) The authors need to delve deeper into the implications of the findings and what the data mean when describing previous studies, especially when the conclusions were clearly different; and c) This manuscript has grammatical and formatting errors throughout and needs extensive editing. Below are some examples of the areas that require improvements:

  1. From the Methods section and Figure 1, I can’t really tell how rigorous the literature screening process was. The description is way too vague. A significant amount of details needs to be added for the discussion to be credible. What makes a record ineligible? What are the automation tools? How were the 454 records excluded? What is the difference between “records” and “reports”? What are the eligibility criteria when assessing the 149 reports?
  2. Write out the full name when an acronym was first introduced in the article. For example, explain what CSM, RNU, RFS, and CSS stand for when they first show up in Abstract. Similarly, explain what pN+ means when it first shows up in section “Radiographic Predictive characteristics”.
  3. Figure 2 needs a lot more specifics. For patients with no clear landmarks of muscle-invasive or locally advanced disease, do the authors recommend LND only when all four predicative factors are present?
  4. Briefly explain what cHG and pHG tumors are in the Introduction.
  5. The Conclusion section doesn’t properly summarize the manuscript. It doesn’t even include the authors’ recommendation.
  6. In Section “Radiographic Predictive characteristics”, can the authors offer some explanation for why the two previous studies reached different conclusions on the predicative utility of hydronephrosis?
  7. How can the authors reconcile the different effects LND has on survival presented by two previous studies described in lines 248-256?
  8. Here is an example of language errors: Recent efforts to combined imaging and ureteroscopic variables to accurately identifiy ≥ pT2 disease have been made. “combined” should be “combine”. “identify” is misspelled.
  9. The formatting for section headings in Discussion needs to be improved. For example, headline “Radiographic predictive characteristics” should be preceded by “3.1.2” instead of “3.2.2”. Similarly, “Combined radiographic and ureteroscopic Variables” should be preceded by “3.1.3” instead of “3.3.3”.

Author Response

Response letter

Thanks for all your comments and insights, we certainly believe made our paper better in every way. The following are the Changes we made point by point:

  1. From the Methods section and Figure 1, I can’t really tell how rigorous the literature screening process was. The description is way too vague. A significant amount of details needs to be added for the discussion to be credible. What makes a record ineligible? What are the automation tools? How were the 454 records excluded? What is the difference between “records” and “reports”? What are the eligibility criteria when assessing the 149 reports?

Response/ We explained the inclusion and exclusion criteria, and what made a record ineligible, modified the PRISMA diagram, so it is easier to understand and standardized the term “records” replacing all the previously mentioned “reports”with it.

  1. Write out the full name when an acronym was first introduced in the article. For example, explain what CSM, RNU, RFS, and CSS stand for when they first show up in Abstract. Similarly, explain what pN+ means when it first shows up in section “Radiographic Predictive characteristics”.

Response/ We reviewed the whole manuscript and before every abbreviation, we made sure the full name was written when first introduced in the article. We explained what pN+ means the first time it was mentioned in the article.

  1. Figure 2 needs a lot more specifics. For patients with no clear landmarks of muscle-invasive or locally advanced disease, do the authors recommend LND only when all four predicative factors are present?

Response/Figure 2 legend, was entirely modified and we gave more specifics of every picture, above the figure we added a new paragraph explaining in detail the indication for LND in our clinical practice.

  1. Briefly explain what cHG and pHG tumors are in the Introduction.

Response/ We explained briefly what cHG and pHG tumors were in the introduction.

  1. The Conclusion section doesn’t properly summarize the manuscript. It doesn’t even include the authors’ recommendation.

Response/ We modified and re-written the conclusions and included our own recommendations for LND and which predictive factors we used in our clinical practice.

  1. In Section “Radiographic Predictive characteristics”, can the authors offer some explanation for why the two previous studies reached different conclusions on the predicative utility of hydronephrosis?

Response/ We gave the following explanation: These findings are in contrast to previous findings in which hydronephrosis tended to overstaged 80% of the tumors[11]. The latter was a study conducted between 1984 and 1995 and included 31 patients. The technological improvements of the cross-sectional imaging techniques in the upcoming years after the study, as well as the limited number of patients, might explain the difference between the results of this study and Messer´s et al results.

  1. How can the authors reconcile the different effects LND has on survival presented by two previous studies described in lines 248-256?

Response/We couldnt address this topic, given that the paragraph mentioned two studies that determined that LND improved CSS. In patients with node-negative disease (cN0), Kondo et al, reported that 42.5% of tumors with cT1 or lower, are upstaged to a clinical stage ≥ pT2 and would lose the benefit of LND, if LND is omitted. This makes difficult to trust conventional predictive factors for muscle invasive UTUC[20]. therefore, it is difficult to use preoperative factors for ≥ pT2, and many experts in the field, as well as EAU guidelines have made their position clear, stating that performing LND in all patients who undergo RNU appears to be reasonable and justified, except for those with severe comorbidities. Lughezzani et al, performed a population-based analysis using the SEER database, including 2824 patients treated with RNU between 1988 and 2004. They reported a 5-year CSS of 81.2% for pN0 and 77.8% for pNx

  1. Here is an example of language errors: Recent efforts to combined imaging and ureteroscopic variables to accurately identifiy ≥ pT2 disease have been made. “combined” should be “combine”. “identify” is misspelled.

Response/ We made the aforementioned corrections and made a native english speaker to reviewed the text in its full length.

  1. The formatting for section headings in Discussion needs to be improved. For example, headline “Radiographic predictive characteristics” should be preceded by “3.1.2” instead of “3.2.2”. Similarly, “Combined radiographic and ureteroscopic Variables” should be preceded by “3.1.3” instead of “3.3.3”

Response/ We reformatted the headlines in the discussion section.

As a review article, the paper adds to the pool of knowledge and nicely summarises the subject of upper tract tumours and management. The authors described appropriate use if search engine and the authors described the methodology clearly. The final conclusions are appropriate. 

There is multiple grammar and punctuation mistakes that require correction. For example; 

Title; Should we offered .... 

In the introduction: Recent efforts to combined ...

Advise revising the punctuations as well. There is excessive use of comma to break sentences. Some are not necessary and some should be separated with fullstops. 

Response/ We made the aforementioned corrections, and reviewed all the punctuation as well avoiding excessive use of comma to break sentences.

Reviewer 2 Report

As a review article, the paper adds to the pool of knowledge and nicely summarises the subject of upper tract tumours and management. The authors described appropriate use if search engine and the authors described the methodology clearly. The final conclusions are appropriate. 

There is multiple grammar and punctuation mistakes that require correction. For example; 

Title; Should we offered .... 

In the introduction: Recent efforts to combined ...

Advise revising the punctuations as well. There is excessive use of comma to break sentences. Some are not necessary and some should be separated with fullstops. 

Round 2

Reviewer 1 Report

The revision demonstrates minor improvements compared to the last version but still didn't quite address my questions and concerns. I noticed that the authors took very little time to turn in the revised version and as a result was not thorough in their edits. I don't believe the manuscript in its current form meets publication standards. 
In my last review, I suggested that the manuscript be improved in three general areas and listed nine examples that fall into these three areas. What needs to be improved is not limited to the nine examples. However, in the response letter, the authors didn’t even mention the three areas I pointed out and, in my opinion, only addressed the nine examples I listed somewhat superficially. The authors need to work on improvements in these three areas throughout the article. For example, grammar errors are still prevalent, and the authors need to articulate what was done specifically in each step of the literature screening process with more details. Due to these considerations, the manuscript still requires major revision to meet publication standards.
To add to my comment in the previous review, the headline “Predictors of Muscle Invasive UTUC” should be preceded by “3.1.1”.

Author Response

Response letter

Thanks for all your comments and insights. The following are the Changes we made for the second revision:

The revision demonstrates minor improvements compared to the last version but still didn't quite address my questions and concerns. I noticed that the authors took very little time to turn in the revised version and as a result was not thorough in their edits. I don't believe the manuscript in its current form meets publication standards.  In my last review, I suggested that the manuscript be improved in three general areas and listed nine examples that fall into these three areas. What needs to be improved is not limited to the nine examples. However, in the response letter, the authors didn’t even mention the three areas I pointed out and, in my opinion, only addressed the nine examples I listed somewhat superficially. The authors need to work on improvements in these three areas throughout the article. For example, grammar errors are still prevalent, and the authors need to articulate what was done specifically in each step of the literature screening process with more details. Due to these considerations, the manuscript still requires major revision to meet publication standards.
To add to my comment in the previous review, the headline “Predictors of Muscle Invasive UTUC” should be preceded by “3.1.1”.

  1. the article in general needs a lot more specifics and details, especially when it comes to recommendation, description of processes, terminology, and acronyms

Answer/ We tried to overcome this issue by adding more evidence to the already written parapraphs of predictive factor of muscle invasive or non-organ confined UTUC and specially to the last paragraphs of the discussion involving LND.  We extensively reviewed all terminology, acronyms, and made sure they were correct, and were described properly when first introduced in the manuscript. All changes were tracked in MS word and we hope you can easily see them.

  1. The authors need to delve deeper into the implications of the findings and what the data mean when describing previous studies, especially when the conclusions were clearly different

Answer/ In order to successfully fulfill this requirement we added at the end of each important paragraph, our opinion, and how those papers might have changed our own clinical practice, and whichi predictive factors, what anatomical based-templates for LND, parameters for surgical adequacy, LNY, etc.. We have adopted in our practice in recent years.  All changes were tracked in MS word and we hope you can easily see them.

  1. This manuscript has grammatical and formatting errors throughout and needs extensive editing.

Answer/ The article was extensively reviewed, and many grammatical errors were found at the second revision and appropriately corrected. The article was reviewed by three different English native-speakers and was approved by then, before sending it back to you. We carried on extensive editing not only of grammatical errors but we change in some phrases the style of writing, and deleted some expressions and some sentences we considered unnecessary. All changes can be easily seen because we tracked all changes in MS word.

  1. From the Methods section and Figure 1, I can’t really tell how rigorous the literature screening process was. The description is way too vague. A significant amount of details needs to be added for the discussion to be credible. What makes a record ineligible? What are the automation tools? How were the 454 records excluded? What is the difference between “records” and “reports”? What are the eligibility criteria when assessing the 149 reports?

Response/ We explained the inclusion and exclusion criteria, and what made a record ineligible, modified the PRISMA diagram, so it is easier to understand and standardized the term “records” replacing all the previously mentioned “reports”with it.

  1. Write out the full name when an acronym was first introduced in the article. For example, explain what CSM, RNU, RFS, and CSS stand for when they first show up in Abstract. Similarly, explain what pN+ means when it first shows up in section “Radiographic Predictive characteristics”.

Response/ We reviewed the whole manuscript and before every abbreviation, we made sure the full name was written when first introduced in the article. We explained what pN+ means the first time it was mentioned in the article.

  1. Figure 2 needs a lot more specifics. For patients with no clear landmarks of muscle-invasive or locally advanced disease, do the authors recommend LND only when all four predicative factors are present?

Response/Figure 2 legend, was entirely modified and we gave more specifics of every picture, above the figure we added a new paragraph explaining in detail the indication for LND in our clinical practice.

  1. Briefly explain what cHG and pHG tumors are in the Introduction.

Response/ We explained briefly what cHG and pHG tumors were in the introduction.

  1. The Conclusion section doesn’t properly summarize the manuscript. It doesn’t even include the authors’ recommendation.

Response/ We modified and re-written the conclusions and included our own recommendations for LND and which predictive factors we used in our clinical practice.

  1. In Section “Radiographic Predictive characteristics”, can the authors offer some explanation for why the two previous studies reached different conclusions on the predicative utility of hydronephrosis?

Response/ We gave the following explanation: These findings are in contrast to previous findings in which hydronephrosis tended to overstaged 80% of the tumors[11]. The latter was a study conducted between 1984 and 1995 and included 31 patients. The technological improvements of the cross-sectional imaging techniques in the upcoming years after the study, as well as the limited number of patients, might explain the difference between the results of this study and Messer´s et al results.

  1. How can the authors reconcile the different effects LND has on survival presented by two previous studies described in lines 248-256?

Response/We couldnt address this topic, given that the paragraph mentioned two studies that determined that LND improved CSS. In patients with node-negative disease (cN0), Kondo et al, reported that 42.5% of tumors with cT1 or lower, are upstaged to a clinical stage ≥ pT2 and would lose the benefit of LND, if LND is omitted. This makes difficult to trust conventional predictive factors for muscle invasive UTUC[20]. therefore, it is difficult to use preoperative factors for ≥ pT2, and many experts in the field, as well as EAU guidelines have made their position clear, stating that performing LND in all patients who undergo RNU appears to be reasonable and justified, except for those with severe comorbidities. Lughezzani et al, performed a population-based analysis using the SEER database, including 2824 patients treated with RNU between 1988 and 2004. They reported a 5-year CSS of 81.2% for pN0 and 77.8% for pNx

  1. Here is an example of language errors: Recent efforts to combined imaging and ureteroscopic variables to accurately identifiy ≥ pT2 disease have been made. “combined” should be “combine”. “identify” is misspelled.

Response/ We made the aforementioned corrections and made a native english speaker to reviewed the text in its full length.

  1. The formatting for section headings in Discussion needs to be improved. For example, headline “Radiographic predictive characteristics” should be preceded by “3.1.2” instead of “3.2.2”. Similarly, “Combined radiographic and ureteroscopic Variables” should be preceded by “3.1.3” instead of “3.3.3”

Response/ We reformatted the headlines in the discussion section.

Round 3

Reviewer 1 Report

  1. The recommendation (line 1556-1560) is confusing and self-contradictory. Should LND be performed if the patient has cHG, cT1 at URS biopsy AND radiographic signs of renal sinus fat? Or only one out of the two variables is sufficient to perform LND? Also, in the following statement “If one of the aforementioned variables is present at preoperative evaluation the risk of losing the potential benefit of LND is too high, and should be avoided whenever possible”, the authors are recommending the exact opposite. This contradiction must be addressed.
  2. The authors still need to significantly improve the quality of the language in this manuscript. I recommend hiring a professional English editor specialized in life sciences. One area I noticed is that the current version has missing commas. For example, in line 1839, there should be a comma after “on the other hand”. I also recommend the use of Oxford comma, such as adding a comma after “to whom” in the title.

Author Response

Response letter

Thanks for all your comments and insights. The following are the Changes we made for the second revision:

  • The recommendation (line 1556-1560) is confusing and self-contradictory. Should LND be performed if the patient has cHG, cT1 at URS biopsy AND radiographic signs of renal sinus fat? Or only one out of the two variables is sufficient to perform LND? Also, in the following statement “If one of the aforementioned variables is present at preoperative evaluation the risk of losing the potential benefit of LND is too high, and should be avoided whenever possible”, the authors are recommending the exact opposite. This contradiction must be addressed.

Answer/  Thanks for this correction, it was an honest mistake and we have addressed it, and corrected it as you can see in the following paragraph

“We recommend performing LND at the time of RNU if a patient has muscle-invasive disease or in a patient with any of the following characteristics: ipsilateral hydronephrosis, cHG, cT1 at URS biopsy or radiographic signs of renal sinus fat or periureteric fat invasion. If any of the variables is present at preoperative evaluation, the risk of losing the potential benefit of LND is too high, therefore, all these patients should undergo LND.”

  • The authors still need to significantly improve the quality of the language in this manuscript. I recommend hiring a professional English editor specialized in life sciences. One area I noticed is that the current version has missing commas. For example, in line 1839, there should be a comma after “on the other hand”. I also recommend the use of Oxford comma, such as adding a comma after “to whom” in the title.

Answer/ We followed the reviewer recommendations and added a come after “ on the other hand” and also we used the serial comma after “to whom”. Other examples of the changes that were made are shown below:

We believe with these templates, all the regions in which LN metastasis could be found are dissected

On the other hand, some experts could argue that with the results of the POUT trial

Why, to Whom, and How Should We Perform Lymph Node Dissection?

The aim of this article is to present a detailed revision on preoperative predictors for muscle-invasive UTUC, locally advanced or advanced UTUC, as well as current indications, technique variations,

medical subject headings (MeSh), Emtree language, Decs, and text words related. We searched through MEDLINE (OVID), EMBASE (Scopus), LILACS, and

recurrence (EUR), locally advanced disease, or high risk UTUC

We try to avoid LND in patients with cLG, cTa, and no ipsilateral hydronephrosis if the patient is expected

for muscle-invasive or locally advanced UTUC, as well as analyzing current indications, technique variations, and the reasons why a LND should be offered to these patients.

prediction tools, and prediction models for risk-stratification.

We included reviews, systematic reviews, and primary studies.

Decs, and text words related. We searched through MEDLINE (OVID), EMBASE (Scopus), LILACS, and the Cochrane Central Register of Controlled Trials (CENTRAL) from inception to May 2021.

the search, conferences, thesis databases, Open Grey, Google scholar, and clinicaltrials.gov

extra urothelial recurrence (EUR), locally advanced disease, or high risk UTUC

We changed citology for cytology

reported a study of 172 patients, 54% with ipsilateral hydronephrosis, 43% with cHG on URS biopsy, and 80%

On multivariate analysis, ipsilateral hydronephrosis

These findings suggested that pNx was significantly associated with worse prognosis than

cT1 at URS biopsy or radiographic signs of renal sinus fat or periureteric fat invasion. If any of the variables are present at

LN metastases to the hilum, paracaval, retrocaval and interaortocaval regions in 22,1%, 44.1%, 10.3%, and 20.6%, respectively. Left side renal pelvis tumors metastasized to the hilum, in 53% of the cases, paraaortic, interaortocaval, common iliac, and aortic

The 4 patients with distal ureter tumors had LN metastases to the para-aortic, common iliac, external iliac, and internal iliac stations in 33.3%, 33.3%, 16.7%, and 16.7% of cases, respectively

For example, if a paracaval LND is performed at least 8 lymph nodes on average should

A LNY of at least 8 LNs, has been shown to predict cancer specific mortality and

In our own practice, we perform LND in patients